# Berberine Derivatives Suppress Cellular Proliferation and Tumorigenesis In Vitro in Human Non-Small-Cell Lung Cancer Cells

**DOI:** 10.3390/ijms21124218

**Published:** 2020-06-13

**Authors:** Jia-Ming Chang, Kam-Hong Kam, Wen-Ying Chao, Pei-Wen Zhao, Shu-Hsin Chen, Hui-Chen Chung, Yi-Zhen Li, Jin-Yi Wu, Ying-Ray Lee

**Affiliations:** 1Department of Surgery, Division of Thoracic Surgery, Ditmanson Medical Foundation Chiayi Christian Hospital, Chiayi City 60002, Taiwan; 06016@cych.org.tw (J.-M.C.); 07137@cych.org.tw (K.-H.K.); 2Department of Physical Therapy, College of Medical and Health Science, Asia University, Taichung 41354, Taiwan; 3Department of Medical Research, Ditmanson Medical Foundation Chiayi Christian Hospital, Chiayi City 60002, Taiwan; s1030722@alumni.ncyu.edu.tw (P.-W.Z.); cych10472@gmail.com (S.-H.C.); hueijenjung@gmail.com (H.-C.C.); 10862@cych.org.tw (Y.-Z.L.); 4Department of Nursing, Min-Hwei College of Health Care Management, Tainan 73658, Taiwan; april@mail.mhchcm.edu.tw; 5Department of Microbiology, Immunology and Biopharmaceuticals, College of Life Sciences, National Chiayi University, Chiayi 60004, Taiwan

**Keywords:** berberine derivatives, non-small-cell lung cancers, anti-cancers

## Abstract

Lung cancer is the leading cause of death in the world, and the most common type of lung cancer is non-small-cell lung cancer (NSCLC), accounting for 85% of lung cancer. Patients with NSCLC, when detected, are mostly in a metastatic stage, and over half of patients diagnosed with NSCLC die within one year after diagnosis; the 5-year survival rate is 24%. However, in patients with metastatic NSCLC, the 5-year survival rate is 6%. Therefore, development of a new therapeutic agent or strategy is urgent for NSCLCs. Berberine has been illustrated to be a therapeutic agent of NSCLC. In the present study, we synthesized six derivatives of berberine, and the anti-NSCLC activity of these agents was examined. Some of them exert increasing proliferation inhibition comparing with berberine. Further studies demonstrated that two of the most effective agents, 9-*O*-decylberberrubine bromide (B6) and 9-*O*-dodecylberberrubine bromide (B7), performed cell cycle regulation, in-vitro tumorigenesis inhibition and autophagic flux blocking, but not induction of cellular apoptosis in NSCLC cells. Moreover, B6 and B7 were determined to be green fluorescent and could be penetrated and localized in cellular mitochondria. Herein, B6 and B7, the berberine derivatives we synthesized, revealed better anti-NSCLC activity with berberine and may be used as therapeutic candidates for the treatment of NSCLCs.

## 1. Introduction

Lung cancer is the second-most common cancer and the leading cause of cancer-associated deaths for both men and women. The 5-year survival is around 17.8%, and, when people are diagnosed with lung cancer, over half of them die within one year [1]. Small-cell lung cancer and non-small-cell lung cancer (NSCLC) are two main subtypes of lung cancer. Among them, around 85% of lung patients are diagnosed with NSCLC [1]. Patients with NSCLC are recommended to remove the tumor with surgery. Moreover, patients with stage II and IIIA NSCLC usually receive adjuvant therapies, including radiation, chemotherapy, or targeted therapy. Therefore, there are lots of chemotherapeutic agents and targeted agents to guide therapy. However, the need to constantly pursue novel therapeutic agents or new treatment strategies to improve overall survival for NSCLC is pressing.

Many natural compounds from herbal extracts are proven to exert anti-cancer ability. Among them, several compounds have been developed as clinical chemotherapeutic agents for various human cancers. Berberine, an isoquinoline alkaloid, is isolated from several herbs, including *Coptis chinenesis, Berberis aristata, Mahonia aquifolium, Phellodendron amurense*, etc., and has potent pharmacological effects including anti-inflammation, anti-oxidative, anti-bacteria, anti-diabetes, anti-cardiovascular, antipsychotic and anti-cancer [2]. Numbers studies have illustrated that berberine exhibits anti-cancer activity in various human cancers [3].

Berberine has anti-NSCLC effects in vitro and in vivo through proliferation inhibition, cell cycle arrest, apoptosis induction and tumorigenesis inhibition, as has been demonstrated [4,5]. These reports suggest that berberine is a potential therapeutic agent for NSCLCs. Previously, we synthesised a series of berberine derivatives, including 9-*O*-alkyl-, 13-alkyl- and 13-*O*-alkylberberine bromide, and evaluated their potential anti-cancer ability in human hepatoma, colon cancer and bladder cancer cell lines [6,7]. We demonstrated that these compounds exhibit stronger proliferation inhibition properties than berberine [6,7]. However, the mechanisms of these berberine derivatives against human hepatoma, colon cancer and bladder cancers are still unclear. Moreover, the anti-tumor ability of berberine derivatives on NSCLCs is also under determination. Herein, we evaluate the anti-NSCLC activity of berberine, berberrubine, and synthesized 9-position substituted berberine derivatives in NSCLC cell lines. Moreover, the underlying anti-NSCLC mechanisms of these compounds were also determined.

## 2. Results

### 2.1. The Effect of Cellular Proliferation of Berberine and Its Derivatives in Non-Small-Cell Lung Cancer Cells

Berberine (B1), berberrubine (B2) and 9-position substituted berberrubine derivatives (B3 to B7; Figure 1) were used to evaluate their biologic activity in human non-small-cell lung cancer cell lines, A549, H23 and H1435 cells, and the cellular viability of these cells was observed (Figure 2). All of the compounds represented growth inhibition activity in NSCLC cells in a dose- and time-dependent manner (Figure 2 and Table 1). Among them, berberine (B1) and berberrubine (B2) exerted a very weak suppression of cell proliferation (Figure 2). However, 9-*O*-hexylberberrubine bromide (B4), 9-*O*-octylberberrubine bromide (B5), 9-*O*-decylberberrubine bromide (B6), and 9-*O*-dodecylberberrubine bromide (B7) had better proliferation inhibition ability for NSCLCs (Figure 2). Notably, B6 and B7 showed the best proliferation inhibition of NSCLC cells among these compounds. Moreover, A549 and H1435 cells were sensitive to B6 and B7 treatment (Table 1) compared with H23 cells. We produced the berberine derivatives B2 to B7 and demonstrated that B6 and B7 could significantly elevate proliferation inhibition activity in NSCLC cells.

### 2.2. 9-O-Decylberberrubine Bromide (B6) and 9-O-Dodecylberberrubine Bromide (B7) Cannot Induce Significant Cellular Apoptosis in NSCLC Cells

B6 and B7 had excellent growth inhibition activity compared with berberine in A549 and H1435 cells (Figure 2 and Table 1). We further evaluated whether B6 and B7 could induce cellular apoptosis in A549 and H1435 cells. After incubation with B6 and B7, the cellular phenomena were determined under microscope, and no apoptotic cells were significantly found (Figure 3). To confirm this finding, Taxol was used as a positive control to induce cellular apoptosis in A549 and H1435 cells, and the profiles of PARP and caspase-3 were examined with western blotting. We found that Taxol could elevate the activation of caspase-3 and PARP and induced cellular apoptosis (Figure 3). However, there was no significant activation of caspase-3 and PARP in A549 or H1435 cells under B6 and B7 treatment (Figure 3). These data demonstrated that both B6 and B7 cannot induce cellular apoptosis in A549 or H1435 cells.

### 2.3. 9-O-Decylberberrubine Bromide (B6) and 9-O-Dodecylberberrubine Bromide (B7)-Induced Cell Cycle Arrest in NSCLC Cells

To address the reason for the growth inhibition by B6 and B7 in A549 and H1435 cells, cell cycle regulation was verified in the cells after incubation with B6 and B7. Both A549 and H1435 exhibited cell cycle arrest at the G0/G1 phase under B6 and B7 treatment, and this phenomenon was associated with incubation concentrations (Figure 4). We also confirmed the expressions of the cell cycle regulation proteins CDK2, CDK4 and p21 in the cells under B6 and B7 treatment. Figure 4 illustrates that the expressions of CDK2 and CDK4 were increased and reduced expression of p21 in the cells after B6 or B7 treatment. These results demonstrate that B6 and B7 can induce G0/G1 arrest in NSCLC cells.

### 2.4. 9-O-Decylberberrubine Bromide (B6) and 9-O-Dodecylberberrubine Bromide (B7) Inhibited In-Vitro Tumorigenesis

To determine the anti-tumorigenesis ability of B6 and B7, an in-vitro colony formation assay was used. A549 and H1435 cells were incubated with B6 or B7 in various dosages, and colony formation was observed and counted after staining with crystal violet. We demonstrated that both B6 and B7 could significantly reduce colony formation in A549 and H1435 cells in a concentration-dependent manner (Figure 5). This result demonstrated that B6 and B7 could be effective anti-tumorigenesis agents in NSCLC cells.

### 2.5. 9-O-Decylberberrubine Bromide (B6) and 9-O-Dodecylberberrubine Bromide (B7) Modulated Autophagy in NSCLC Cells

Autophagy is a self-degradative mechanism, which can disassemble dysfunctional or unnecessary elements in the cells, and accordingly, maintain homeostasis and intracellular energy balance [8]. Autophagy has been reported to play a dual role in cancer. It can promote cancer growth and survival by maintaining cellular energy production and eliminating stress; however, it has also been demonstrated to be a therapeutic strategy against cancer [9]. Therefore, we also evaluated autophagy regulation in A549 and H1435 cells under B6 or B7 treatment. LC3-II elevation was illustrated in both A549 and H1435 cells under B6 and B7 treatment in a concentration-dependent manner (Figure 6A,B), which suggested that autophagy induction or autophagic flux suppression occurred in the cells under B6 and B7 treatment. Further study showed that B6 and B7 increased LC3-II performance associated with incubation time (Figure 6C), suggesting that autophagic flux was suppressed in the cells after incubation with B6 and B7. To confirm this suggestion, a pmRFP-EGFP-LC3 plasmid was transfected into A549 cells, and the autophagosome and autolysosome puncta were examined with confocal microscopy. Autophagosome and autolysosome induction and enhanced autophagic flux were illustrated in the cells under rapamycin treatment (Figure 6D). Moreover, chloroquine treatment could block endogenous autophagic flux in A549 cells (Figure 6D). However, both B6 and B7 could suppress endogenous and rapamycin-induced autophagic flux (Figure 6D), demonstrating that B6 and B7 act as autophagic flux blockers. Further study showed that blocking autophagy with 3-MA could partially reverse B6 and B7, causing cell death (Figure 6E). Additionally, by enhancing autophagy with rapamycin in B6- or B7-treated cells, cell viability was reduced compared with cells incubated with B6 or B7 only (Figure 6E). We demonstrated that B6 and B7 could modulate cellular autophagy in NSCLC cells, and enhanced cellular autophagy in B6/B7-treated cells was suggested to elevate anti-cancer behavior.

### 2.6. 9-O-Decylberberrubine Bromide (B6) and 9-O-Dodecylberberrubine Bromide (B7) Localized in Cellular Mitochondria and Emitted Green Fluorescence

Our previous report demonstrated the absorption and emission spectra of 9-*O*-dodecylberberrubine bromide (B7) are 420 nm and 529 nm in the cells [6]. In the present study, we confirm this phenomenon for B6 and B7 in A549 and H1435 cells after incubation. A significantly green fluorescence (GFP) emission was detected in the cells under B6 and B7 treatment (Figure 7A and 7B). Moreover, we also demonstrated that these GFP signals were co-localized with cellular mitochondria in both A549 and H1435 cells (Figure 7A,B). These results demonstrate that B6 and B7 could emit green fluorescence and could be taken into the cells after incubation, and they will be localized on the cellular mitochondria.

## 3. Discussion

NSCLC is the most common type of lung cancer, making up around 85 percent of all lung cancer cases. Chemotherapy and target therapy are recommended therapies for NSCLCs. However, the 5-year survival rate for NSCLC is only 24%. Therefore, development of a novel therapeutic agent or effective strategy to improve overall survival for NSCLC is pressing. In the previous studies, berberine shows anti-tumor activity in various human cancers, including NSCLCs [3,4,5]. Moreover, berberine has been demonstrated to be a safety agent for cancer therapy [10]. In our previous study, we generated a series of compounds from berberine with 9-position-substituted derivatives B2 to B7, which exert better anti-cancer behavior than berberine in human hepatoma, colon and bladder cancer cells [6]. Here, we demonstrated that berberine derivatives, including 9-*O*-hexylberberrubine bromide (B4), 9-*O*-octylberberrubine bromide (B5), 9-*O*-decylberberrubine bromide (B6), and 9-*O*-dodecylberberrubine bromide (B7), have better anti-NSCLC activity than berberine (B1) (Figure 2 and Table 1). Among them, B6 and B7 show the most inhibition of NSCLC cells. However, among A549, H23 and H1435 cells, H23 cells had more resistance than A549 and H1435 cells to B6 and B7 treatment (Table 1). Genetic differences may the reason to explain this result. Among these three cell lines, PTEN mutation has been found in H23 cells [11]. *PTENT* has been considered a predictive biomarker of Akt activation and response to therapies in multiple cancers [12]. In addition, Yoon et al. demonstrate that PTEN mutation might render KRAS mutant cancer cells less sensitive to the treatment of MEK inhibition [11]. Berberine and its derivatives suppress the MEK-ERK signaling pathway, as has been reported in various studies [13,14,15,16,17]. Therefore, whether *PTEN* mutation in H23 cells is the reason for the resistance to B6 and B7 needs further investigation.

A previous study reported that berberine and its derivatives could be taken up by cancer cells, and they could be excited with a wavelength of 420 nm and emit wavelengths of 529 to 531 nm [6]. Moreover, they show photocytotoxicity in hepatoma, colon and bladder cancer cells [6]. We also confirmed that B6 and B7 could emit green fluorescence under excitation with 488 nm (Figure 7). In addition, B6 and B7 were taken up into the cells and localized on the cellular mitochondria (Figure 7). However, cellular apoptosis in B6- and B7-incubated cells was not found (Figure 3), suggesting that B6 and B7 localized on the cellular mitochondria cannot induce mitochondria-dependent apoptosis. Therefore, the biological effect of B6 and B7 binding on the mitochondria is still unclear.

Our previous study demonstrated that an S phase arrest was observed in the cells incubated with berberine and its derivatives after exposure to irradiation (420 nm) [6]. Moreover, cellular apoptosis was elevated under this condition. However, in the present study, B6 and B7 could induce cell cycle arrest at G0/G1 in human NSCLC cells (Figure 4), and no cellular apoptosis was examined (Figure 3). These data suggest that B6 and B7 could exert anti-cancer effect in NSCLCs with non-photo- or photo-cytotoxicity, involving different mechanisms.

Berberine-induced autophagy may be an adaptive antitumor response, however, it was also observed to have a protective role in human tumors [18]. Berberine modulates cellular autophagy in human hepatoma cells, and it can ameliorate the blockade of autophagy flux caused by cholesterol in hepatoma cells [19]. Moreover, berberine induces autophagy and prevents lung carcinogenesis, as has also been reported [18]. In this study, we demonstrated that B6 and B7 could block cellular autophagic flux (Figure 6A–D). Moreover, enhancing cellular autophagy in the cells under B6 or B7 treatment could elevate the antitumor activity of NSCLCs. Berberine regulates autophagy, which contributes to its protective roles against myocardial hypoxia/reoxygenation injury, diabetic nephropathy, neuronal damage, hepatic steatosis, Alzheimer’s disease and pulmonary fibrosis, as well as its anti-viral activity [17,20,21,22,23,24,25]. It is interesting to evaluate whether B6- and B7-mediated autophagic flux regulation can promote favorable activities to berberine on these issues.

In summary, we demonstrated that B6 and B7 had excellent antitumor activity vs. berberine, and they caused growth inhibition and tumorigenesis suppression but not apoptosis induction in NSCLC cells. Moreover, B6 and B7 also modulated cellular autophagy, which contributed to its antitumor activity. They can be taken up into the cells and localized in the mitochondria. In the present study, we generated a series of berberine derivatives, and B6 and B7 could be used as novel anti-NSCLC agents. However, further studies are needed to evaluate the application of these compounds as therapeutics.

## 4. Materials and Methods

### 4.1. Chemical and Reagents

Generic chemicals were purchased from Sigma-Aldrich (MO, USA). Berberine (B1), berberrubine (B2), 9-*O*-butylberberrubine bromide (B3), 9-*O*-hexylberberrubine bromide (B4), 9-*O*-octylberberrubine bromide (B5), 9-*O*-decylberberrubine bromide (B6), and 9-*O*-dodecylberberrubine bromide (B7) were synthesized and provided by Dr. Jin-Yi Wu [6,26]. Primary antibodies, caspase-3 (#9662), PARP (#9542) and p21 (#2947) were purchased from Cell Signaling Technology, Inc. (Beverly, MA, USA); GAPDH (GTX100118) was purchased from GeneTex, Inc. (Irvine, CA, USA); CDK2 (#2351-1) was purchased from Epitomics, Inc. (Burlingame, CA, USA); CDK4 (SC601) was purchased from Santa Cruz Biotechnology, Inc. (Dallas, TX, USA). All the chemicals and biochemicals used in this study were of analytical grade.

### 4.2. Cell Lines and Cell Cultures

The lung adenocarcinoma lines, A549, H23 and H1435, were purchased from the American Type Culture Collection, (Manassas, Virginia, USA) and respectively cultured in DMEM, RPMI and DMEM mixed with F12K (1:1) medium containing 10% fetal bovine serum, l-glutamine (2 mM), streptomycin (100 μg/mL) and penicillin (100 IU/mL) (all from Gibco-Invitrogen Corp., Carlsbad, CA, USA). Cells were incubated at 37 °C in a humidified atmosphere containing 5% CO_2_.

### 4.3. Cellular Viability Assay

Cells were seeded into 96-well plates at a density of 5 × 10^3^ cells per well and grown in the aforementioned medium. After overnight attachment, cells were administrated with control medium (containing 0.01% dimethyl sulfoxide, DMSO) or medium with indicated agents. Cellular viability was observed with the CCK-8 assay kit (Enzo Life Sciences, Farmingdale, NY, USA) after incubation. Three independent assays were performed in this study.

### 4.4. Western Blotting and Immune-Fluorescent Staining

Cells were lysed with M-PERTM protein extraction reagent (Thermo Fisher Scientific Inc., Rockford, IL, USA) containing a 0.1% protease inhibitor cocktail after the treatments. The sample proteins were loaded and separated with sodium dodecyl sulfate–polyacrylamide gel electrophoresis (SDS-PAGE) gels. After transfer with polyvinylidene fluoride (PVDF) membranes, proteins were detected by primary antibodies and secondary antibodies. The detailed experimental procedure has been described in our previous study [27].

To determine the mitochondria, cells were treated with control medium or study medium with compounds, and the mitochondria were stained with Mitotracker Red CMXRos (M7512; Thermo Fisher) for 30 min at 37 °C. The immunofluorescence signal was observed with a laser confocal scanning microscope (LSM800, ZEISS, Germany).

### 4.5. Cell Cycle Distribution

The cell cycle analysis has been described previously [27,28,29,30,31,32]. Cells were cultured in 10-cm dishes overnight and were treated with serum starvation for 24 h before incubation with control medium or medium with study compounds. The cells were stained with propidium iodide (PI) (Sigma-Aldrich) containing RNase for 30 min in the dark at room temperature. The cell cycle profiles were examined by FACScan flow cytometer (Becton Dickinson, San Diego, CA, USA). DNA content was further analyzed by Modfit LT 3.3 software.

### 4.6. Colony Formation Assay

Cells were seeded in 6-well plates (10^3^ cells/well) and incubated at 37 °C in the aforementioned medium. The cells were then treated with control medium or medium with compounds. The colony-forming efficiency of the cells and morphology of the colonies were observed after staining with 10% crystal violet (Sigma-Aldrich) during 12 days post incubation. The numbers of the colonies were counted.

### 4.7. Autophagy Observation

Cells were incubated with control medium or medium with compounds, and the expressions of LC3-I and LC3-II (Medical & Biological Laboratories, Nagoya, Japan), and p62 (sc-28359; Santa Cruz Biotechnology) were examined with western blotting. To observe autophagosome and autolysosome formation in cells treated with B6 or B7, pmRFP-EGFP-LC3 plasmid (purchased from Addgene, Watertown, MA, USA) was transfected with Lipofectamine 2000 (Thermo Scientific, Waltham, MA, USA) according to the manufacturer’s instructions, and the autophagosome (yellow) and autolysosome (red) puncta were illustrated with confocal scanning microscopy (LSM800, ZEISS, Germany).

### 4.8. Statistical Analysis

Data are presented as the mean ± standard error for the indicated number of separate experiments. Statistical differences were analyzed by one-way ANOVA and Fisher’s least significant difference test. Statistical significance was defined as *p* < 0.05 in all tests.

## Figures and Tables

**Figure 1 ijms-21-04218-f001:**
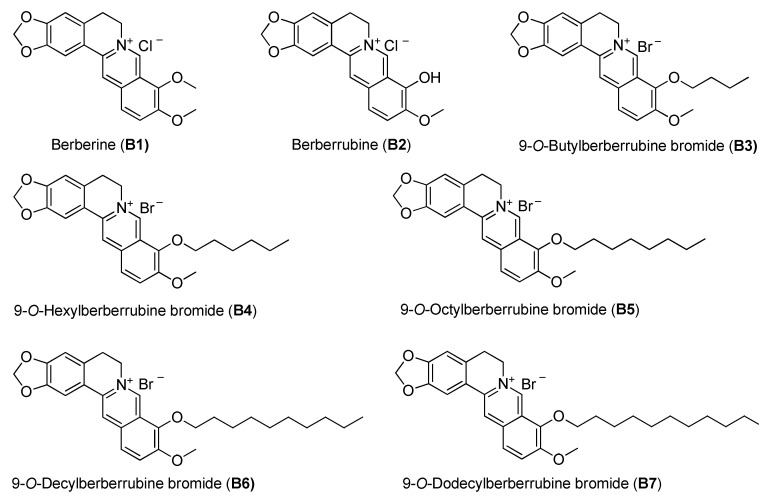
The chemical structures of berberine and its derivatives.

**Figure 2 ijms-21-04218-f002:**
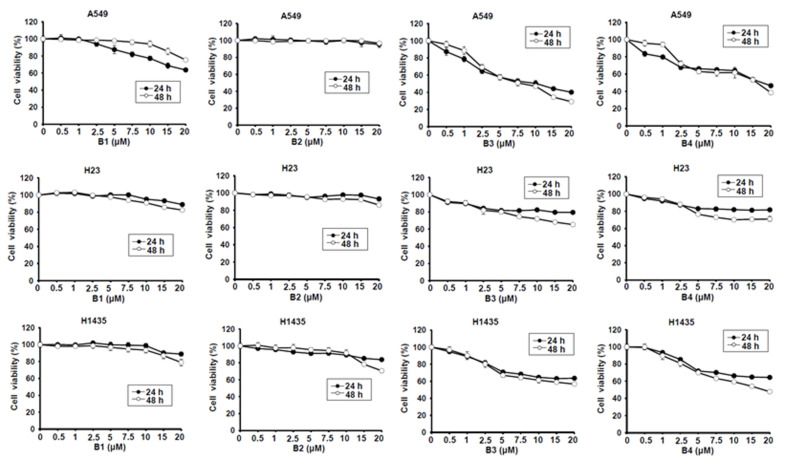
Cellular survival of berberine (B1), berberrubine (B2) and its derivatives B3–B7 in human non-small-cell lung cancer cells. Human non-small-cell lung cancer cell lines, A549, H23 and H1435, were incubated with control medium or various compounds with indicated dosages and the cellular viability was examined at 24 h and 48 h.

**Figure 3 ijms-21-04218-f003:**
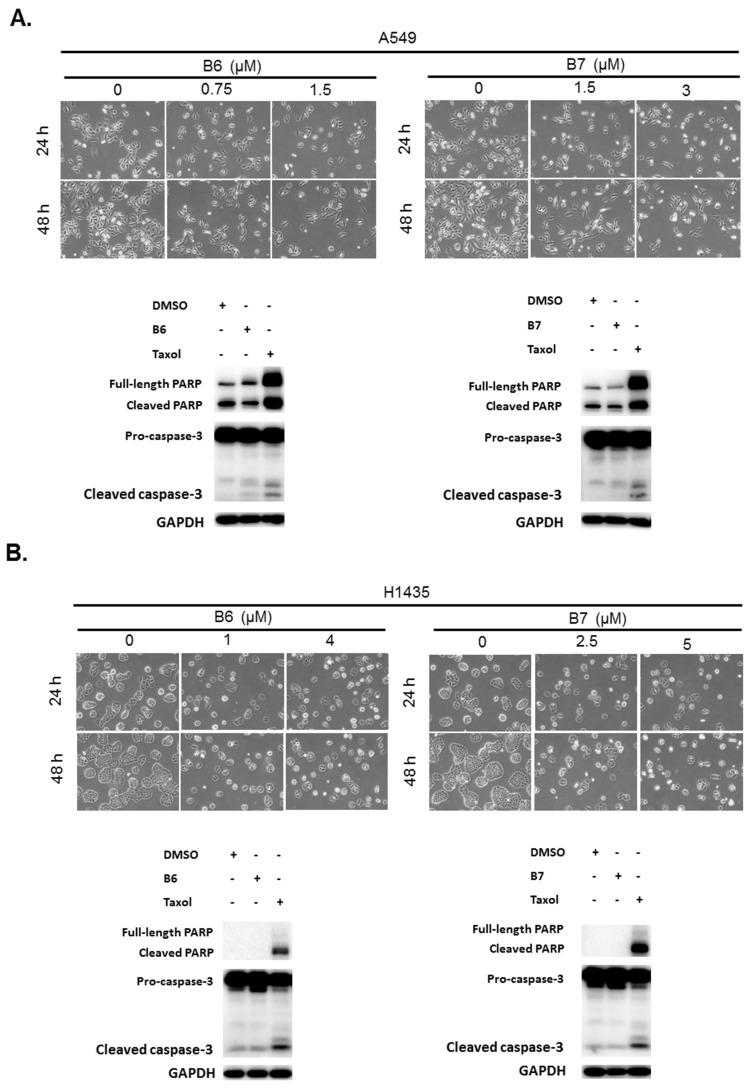
Cellular apoptosis profile in non-small-cell lung cancer cells under B6 and B7 treatment. (**A**) A549 and (**B**) H1435 cells were treated with control medium or B6 and B7, and the cellular morphology was determined under microscopy. Total cell lysate was extracted after 24-h incubation, and the expressions of caspase-3 and PARP were determined by western blotting. DMSO was used as a negative control, and Taxol was used as a positive control. GAPDH was used as a loading control. Three independent experiments were conducted.

**Figure 4 ijms-21-04218-f004:**
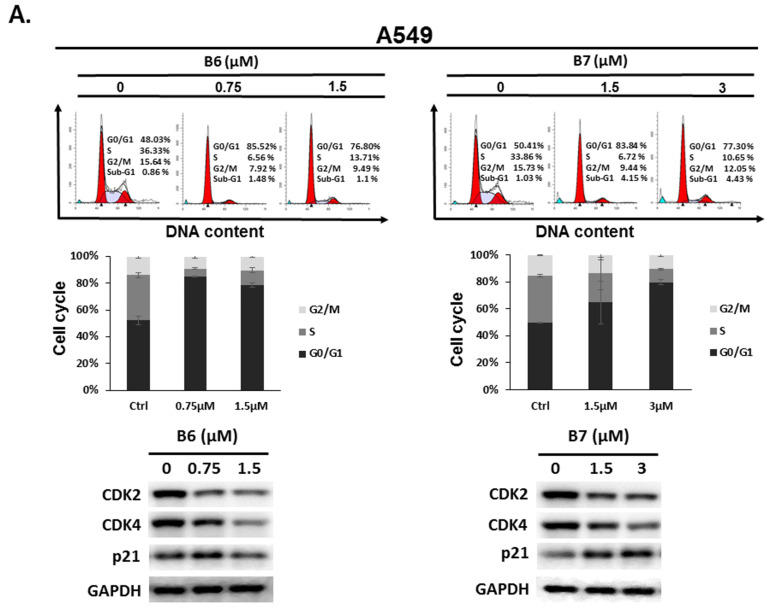
Cell cycle profile for in non-small-cell lung cancer cells after incubation with B6 and B7. (**A**) A549 and (**B**) H1435 cells were incubated with control medium or B6 and B7 for 24 h, and the cell cycle was examined with FACS flow cytometry. Additionally, the total cell lysate was isolated and the expressions of CDK2, CDK4, and p21 were verified with western blotting. DMSO was used as a negative control. GAPDH was used as a loading control. Three independent experiments were conducted.

**Figure 5 ijms-21-04218-f005:**
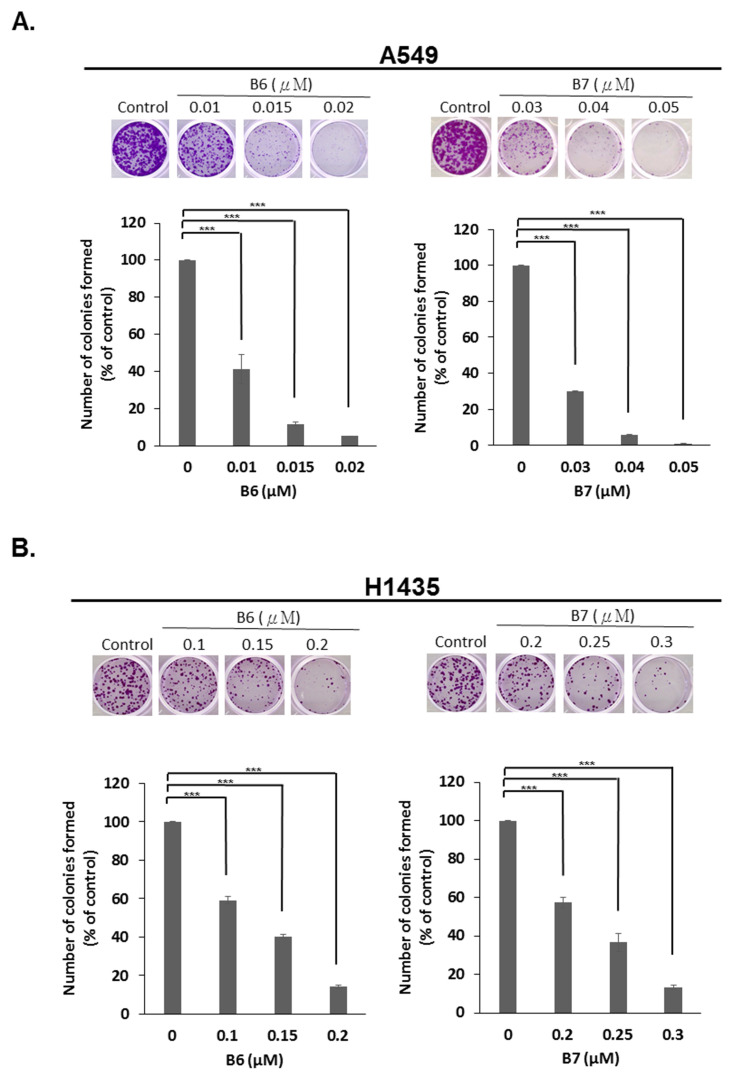
Suppression of in-vitro tumorigenesis in non-small-cell lung cancer cells with B6 and B7. (**A**) A549 and (**B**) H1435 cells were treated with control medium or B6 and B7, and the colonies formation assay was used to evaluate the modulation of in-vitro tumorigenesis ability. GAPDH was used as a loading control. Five independent experiments were conducted. *** means *p* < 0.001.

**Figure 6 ijms-21-04218-f006:**
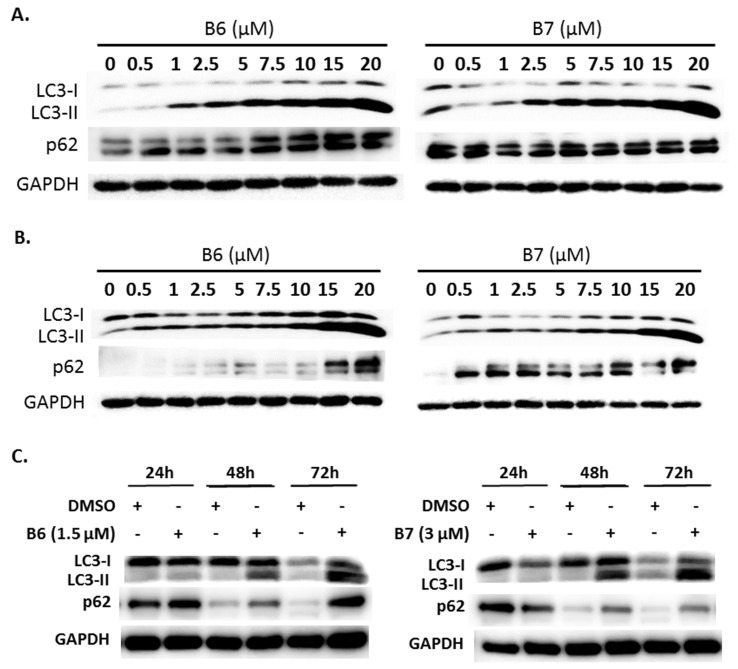
Autophagy regulation of B6 and B7 in non-small-cell lung cancer cells. (**A**) A549 and (**B**) H1435 cells were incubated with control medium or B6 and B7, and the expressions of LC3-I/LC3-II and p62 were observed using western blotting. (**C**) LC3-I/LC3-II and p62 were determined in A549 cells under B6 and B7 treatment with their time course. GAPDH was used as a loading control. Three independent experiments were conducted. (**D**) Autophagic flux determined in A549 cells under B6 and B7 treatment. A549 cells were transfected with pmRFP-EGFP-LC3 and were treated with B6 (1.5 µM), B7 (3 µM), rapamycin (30 µM), and chloroquine (10 µM) for 48 h. The autophagosome (yellow) and autolysosome (red; marked by arrow) puncta were determined under confocal microscopy. DMSO was used as a negative control. (**E**) The cellular viability of A549 cells was examined in the cells’ incubation with B6 or B7 or in combination with 3-MA or rapamycin after 48 h post treatment. 3-MA was an autophagic blocker, and rapamycin was used as an autophagic activator. *** and ^###^ means *p* < 0.001.

**Figure 7 ijms-21-04218-f007:**
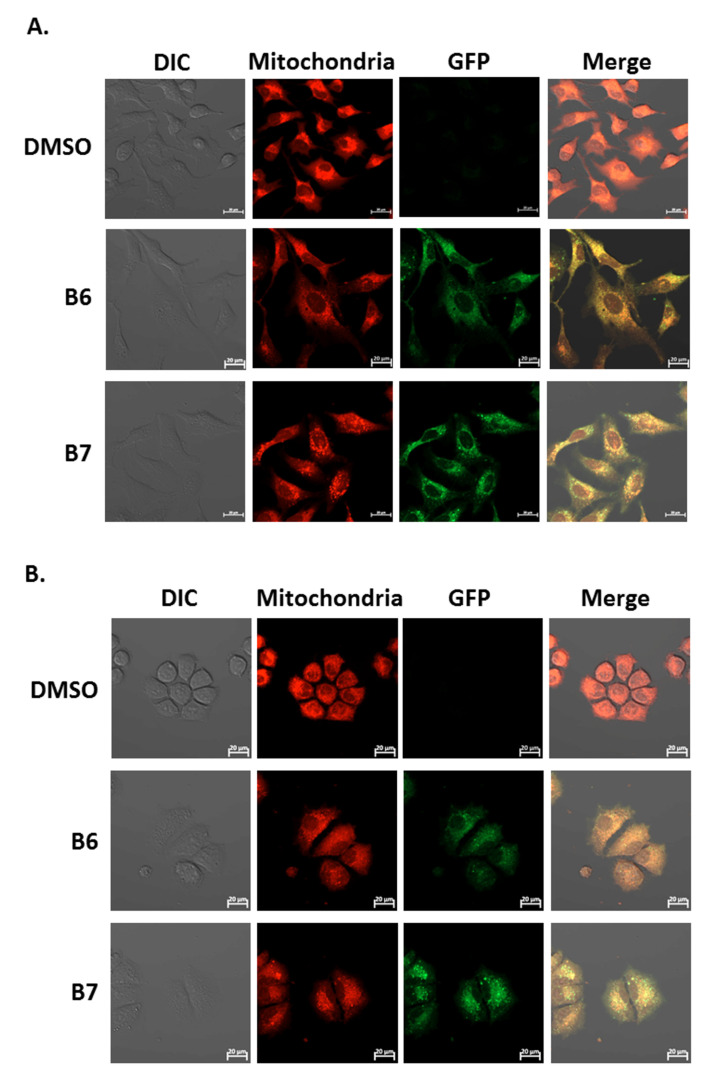
The uptake and localization of B6 and B7 in non-small-cell lung cancer cells. (**A**) A549 and (**B**) H1435 cells were incubated with control medium or B6 and B7 for 24 h, and the mitochondria was stained with mitotracker and determined under confocal microscope. The expression and the localization of B6 and B7 were shown by excitation at 488 nm, and the emission of green fluorescence from B6 and B7 was observed under a confocal microscope.

**Table 1 ijms-21-04218-t001:** The IC_50_ of berberine and its derivatives in human non-small-cell lung cancer cells.

Cell	Time (h)	IC_50_ (μM)						
		B1	B2	B3	B4	B5	B6	B7
A549	24 h	33.3	133.0	10.3	17.6	3.3	1.4	2.8
	48 h	32.5	100.3	8.1	16.1	1.5	0.7	1.4
H23	24 h	65.2	70.2	199.4	237.9	14.4	11.4	10.8
	48 h	69.1	49.1	46.2	118.0	11.6	8.7	8.8
H1435	24 h	67.0	140.7	124.2	64.3	7.7	4.3	5.3
	48 h	57.8	33.4	35.2	18.3	1.6	1.1	2.5

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
