# Peer review of "Berberine Derivatives Suppress Cellular Proliferation and Tumorigenesis In Vitro in Human Non-Small-Cell Lung Cancer Cells"

_ijms, 2020, doi:10.3390/ijms21124218_

Round 1
Reviewer 1 Report
Review of the manuscript ID ijms-830740 "Berberine derivatives suppress cellular proliferation and tumorigenesis in vitro in human non small cell lung cancer cells "
The manuscript describes the investigation of the anti-NSCLC activity of berberine, berberrubine chloride and five of its 9-O-alkyl derivatives. It is shown that 9-O-decyl and 9-O-dodecyl berberrubine bromides B6 and B7 has the best activity.
The manuscript has the following comments:
1) Authors need to redo the abstract. The description of NSCLC in it is unnecessary, it is better to focus on the results achieved. In the abstract (l. 29) it is written that 9 derivatives have been synthesized, but in fact berberine and 6 of its derivatives have been studied. In the abstract, instead of the names B6 and B7, authors must use the nomenclature's names of compounds.
2) Special attention should be given to the names of these compounds due to the complexity of their construction according to the IUPAC nomenclature's rules. The variants used by the authors (for example, 9-O-dodecyl- (l. 62), 9-octylerberine (B5) (l.78, Figure 1, l. 93, and others) are not completely correct. When using the substitution nomenclature and the berberine bromide molecule as the source, the correct name is, for example, "9-Demethyl-9-O-dodecylberberine bromide". When using berberrubine hydrochloride as the initial structure, the correct name is "9-O-dodecylberberubine bromide". Examples of using such nomenclature can be found in [Chem Nat Compd 48, 1047-1053 (2013). https://doi.org/10.1007/s10600-013-0461-z; Russ Chem Bull 68, 1052–1060 (2019). https://doi.org/10.1007/s11172-019-2519-y]. The names of compounds must be corrected throughout the text of the article.
3) The reference to the synthesis of compounds B1-B7 should be provided (reference [7]?) before they are discussed in section 2.1. The same should be done in section 4.1. The link [7] itself is incorrect – there are no pages in it. In addition, these compounds are not new (synthesized for the first time) and it is necessary to provide a reference to their original synthesis.
4) It is written in section 4.6 that "Data are presented as the mean ± standard error", but neither the text nor the tables indicate the standard error.
5) The compound B5 shows results comparable to compound B7 at a time of 48 hours, according to Table 1. It is necessary to explain why the researchers did not consider to study the B5 compound in detail further?
There were listed the comments that were notable from a chemical point of view. The review should also be reviewed by a cell biologist.
These comments are minor, and after correcting them, the article may be recommended for publication.
Author Response
Reviewer 1: (Changes in the manuscript are highlighted in red)
Q1. In the abstract (l. 29) it is written that 9 derivatives have been synthesized, but in fact berberine and 6 of its derivatives have been studied. In the abstract, instead of the names B6 and B7, authors must use the nomenclature's names of compounds.
Reply comment:
Thanks very much for your reminding. It was amended to “In the present study, we synthesized 6 derivatives of berberine, … two of the most effective agents, 9-O-decylberberrubine bromide (B6), and 9-O-dodecylberberrubine bromide (B7),” in the “Abstract” of the revised manuscript.
Q2. Special attention should be given to the names of these compounds due to the complexity of their construction according to the IUPAC nomenclature's rules. The variants used by the authors (for example, 9-O-dodecyl- (l. 62), 9-octylerberine (B5) (l.78, Figure 1, l. 93, and others) are not completely correct. When using the substitution nomenclature and the berberine bromide molecule as the source, the correct name is, for example, "9-Demethyl-9-O-dodecylberberine bromide". When using berberrubine hydrochloride as the initial structure, the correct name is "9-O-dodecylberberubine bromide". Examples of using such nomenclature can be found in [Chem Nat Compd 48, 1047-1053 (2013). https://doi.org/10.1007/s10600-013-0461-z; Russ Chem Bull 68, 1052–1060 (2019). https://doi.org/10.1007/s11172-019-2519-y]. The names of compounds must be corrected throughout the text of the article.
Reply comment:
It was amended the correct names of compounds in the revised manuscript. The names of all compounds was amended to “9-O-butylberberrubine bromide (B3), 9-O-hexylberberrubine bromide (B4), 9-O-octylberberrubine bromide (B5), 9-O-decylberberrubine bromide (B6), and 9-O-dodecylberberrubine bromide (B7)”.
Q3. The reference to the synthesis of compounds B1-B7 should be provided (reference [7]?) before they are discussed in section 2.1. The same should be done in section 4.1. The link [7] itself is incorrect – there are no pages in it. In addition, these compounds are not new (synthesized for the first time) and it is necessary to provide a reference to their original synthesis.
Reply comment:
It was added a new reference in the revised manuscript.
- Kim, S.H.; Lee, S.J.; Lee, J.H.; Sun, W.S.; Kim, J.H. Antimicrobial activity of 9-O-acyl- and 9-O-alkylberberrubine derivatives. Planta Med. 2002, 68, 277–281.
Q4. It is written in section 4.6 that "Data are presented as the mean ± standard error", but neither the text nor the tables indicate the standard error.
Reply comment:
Data are presented as the mean ± standard error in the bar and line charts in Figure 2, 4, 5 and 6.
Q5. The compound B5 shows results comparable to compound B7 at a time of 48 hours, according to Table 1. It is necessary to explain why the researchers did not consider to study the B5 compound in detail further?
Reply comment:
In the present study, the anti-NSCLCs activity of these compounds is B6 > B7 > B5 > B4 > B3 and B1 > B2. Therefore, we only examine the best two agents including B6 and B7 of their anti-cancer activity and the related mechanisms in this study. At present, compound B5 has been tested in the bladder cancer animal models, and another article will be published in the future.
Reviewer 2 Report
An interesting article describing the effects of berberine derivatives. Overall the techniques and conclusions are fine, but the manuscript needs improvements. 1) berberine and its effects are the focus of the whole study, Introduction needs significantly improved explanation what is already known and why this study was undertaken.
2) The importance of lung cancer is well known and it is not necessary to fill the space with information about smoking risk. Please shorten.
3) Some parts do not fully correspond - Data in Table one do not always correspond to data in Figure 2.
4) Standard errors and statistics are needed.
Author Response
Q1. Berberine and its effects are the focus of the whole study, Introduction needs significantly improved explanation what is already known and why this study was undertaken.
Reply comment:
Thanks very much for your reminding. We have illustrated the effects of berberine in the Introduction and Discussion sections. (Marked with yellow)
Q2. The importance of lung cancer is well known and it is not necessary to fill the space with information about smoking risk. Please shorten.
Reply comment:
This part of the narrative has been deleted in the revised manuscript.
Q3. Some parts do not fully correspond - Data in Table one do not always correspond to data in Figure 2.
Reply comment:
Figure 2 showed the proliferation inhibition activity of these compounds on three NSCLC cells with only 0 to 20 µM in 24 and 48 h post-treatment. However, at these conditions, B1, B2, B3 and B4 didn’t exert half maximal inhibitory concentration. Therefore, the IC50 at 24 and 48 h post-treatment of these compounds in Table 1 are determined with large scale dosages and calculated by Graphpad software.
Q4. Standard errors and statistics are needed.
Reply comment:
Data are presented as the mean ± standard error in the bar and line charts in Figure 2, 4, 5 and 6.
Round 2
Reviewer 1 Report
The authors have corrected the shortcomings indicated by them. The article is recommended for publication.
Reviewer 2 Report
The changes significantly improved the quality of the manuscript.